# The Impact of Soil pH on Heavy Metals Uptake and Photosynthesis Efficiency in *Melissa officinalis*, *Taraxacum officinalis*, *Ocimum basilicum*

**DOI:** 10.3390/molecules27154671

**Published:** 2022-07-22

**Authors:** Dorota Adamczyk-Szabela, Wojciech M. Wolf

**Affiliations:** Institute of General and Ecological Chemistry, Lodz University of Technology, Zeromskiego 116, 90-924 Lodz, Poland; wojciech.wolf@p.lodz.pl

**Keywords:** uptake of metals, toxicity, soil reaction, photosynthesis, herbs

## Abstract

Herbs used in medicine should be grown under controlled and standardized conditions. Agricultural practices often induce changes to soil pH, which may affect migration of heavy metals in the environment, their accumulation in plant tissues and the concentration of medicinal ingredients. The aim of this work was to assess the influence of various soil pH on the biological parameters and uptake of manganese, copper and zinc by basil, dandelion and lemon balm. The soil analysis covered pH, organic matter content, bioavailable and total forms of investigated metals in soil. In plants cultivated in soil at pH covering the range 4.7–8.5 the concentrations of Mn, Cu and Zn were analyzed. Their mobility and availability were assessed by bioaccumulation factors, translocation factors and transfer coefficients. The seed germination and subsequent herbs growth were strongly dependent on soil pH for all investigated plant species. Photosynthetic efficiency at different pHs was positively correlated with uptake of Cu and Mn while Zn behaved in a more random way.

## 1. Introduction

Soil pH affects almost all biological and chemical processes which control terrestrial ecosystems. Notably, it alters plant metabolism and biomass yield [1]. In particular, the soil pH significantly influences the solubility, bioavailability and translocation of heavy metals in plant species [2]. The latter interaction has been well documented in numerous studies [3]. In general, the majority of micronutrients are better available to plants cultivated in acidic soils than in neutral or alkaline conditions [4]. On the other hand, soil pH mediates metal toxicity [5]. At high pH, metals tend to form barely soluble phosphates and carbonates. On the contrary, at low pH they are likely to exist in more bioavailable free ionic forms [6]. Their migration is to a large extent mediated by Coulombic forces [7]. In particular, uptake of metal cations from the rhizosphere by the root usually results in the discharge of equivalent number of protons [1].

In these conditions metal cations compete with protons embedded in biological matrices and are involved in diverse interactions [8]. On the other hand, numerous studies emphasize that plants can be significantly affected by the biologically available forms of metals in soil only. The remaining fraction is sequestered or irreversibly bounded to the soil matrix and usually does not interact directly with plants. Notable, both forms of metals can be transformed one into another by changes of the soil pH [9].

Zinc, manganese and copper are essential for plants at low levels and are involved in a number of biochemical reactions. In particular, zinc is a cofactor located in an active site of chloroplastic b carbonic anhydrase (b-CA), which catalyzes rapid, reversible interconversion of HCO_3_^−^ and CO_2_ and supplies RuBisCo with CO_2_ [10]. Additionally, Zn-finger proteins are engaged in photosynthesis where they affect transcription through the site-specific interactions [11].

Manganese is involved in a number of processes which govern plant growth and development. In particular, it initiates photosynthesis by the water-splitting reaction in the photosystem II. It is catalyzed by the cluster Mn_4_O_5_Ca, which supports partition of two water molecules into four electrons, four protons, and molecular O_2_ [12,13].

Copper is a key component of the electron transport chain in photosystem I and acts as a redox center in plastocyanin macromolecule [14,15]. It is also involved in the activity of Cu/Zn-superoxide dismutase (Cu/Zn-SOD), a protein which is localized in the stroma of chloroplasts and is involved in the scavenging of reactive oxygen species [10].

Herbs and spice plants are particularly susceptible to changes in soil, which substantially affects their constituency and properties. Therefore, careful adjustment of growing conditions is crucial as far as medicinal value of herbs is concerned [16]. In this work we investigated three popular herbs of high medicinal value. *Melissa officinalis* L., (lemon balm, bee balm, honey balm) is a perennial herb belonging to the *Lamiaceae* (mint) family. This plant is native to Europe, Central Asia and Iran but now has been disseminated in both Americas and elsewhere [17]. Its leaves contain flavonoids, beneficial volatile compounds, triterpenes, polyphenols and essential oils. *M. officinalis* possesses sedative, antibacterial, antiviral, and antifungal activities. It is an easy herb to grow that can be cultivated in diverse soils and climatic conditions [18].

*Taraxacum officinalis* (dandelion) is a member of the *Asteraceae* family. It is a perennial herb, native throughout the Northern hemisphere, found growing wild in meadows, pastures and waste grounds of temperate zones. Dandelion root and leaf are used widely in Europe for gastrointestinal ailments. Due to the high content of minerals, fiber, vitamins, unsaturated fatty acids, inulin, phytosterols, amino acids [19], sesquiterpenes, triterpenes, phytosterols and phenolic compounds, dandelion is a very popular medicinal plant with valuable properties [20].

*Ocimum basilicum* L. (sweet basil) is cultivated in many countries of the world as an edible, medicinal and industrial plant [21]. It has antibacterial, antioxidant, antifungal and anti-inflammatory properties. Basil contains many valuable ingredients, including rosmarinic and cinnamic acid, flavonoids and anthocyanins, essential oils and saponins [22].

In order to ensure highly efficient crop production, inorganic fertilizers and pesticides are widely applied. They affect diverse components of the environment, with the accumulation of heavy metals in soil and plants being the most prone. Furthermore, herbs and spice plants are particularly susceptible to changes in soil, which substantially affects their constituency and properties. Therefore, careful adjustment of growing conditions is crucial as far as medicinal value of herbs is concerned [16]. This work follows our investigations on how soil pH controls heavy metals uptake by herbs and for the first time shows photosynthetic background of this process.

## 2. Results

### 2.1. Analysis of Soil Used in the Study

The soil used in this study was organic and acidic [23,24,25]. Total manganese, copper and zinc contents demonstrated that it was not contaminated by these metals [26,27] (Table 1a). Bioavailable forms of Mn, Cu and Zn were determined in either initial soil without additives at pH = 6.0 or modified soil samples with pH adjusted to 4.7 and 8.7, respectively (Table 1b). Acidic conditions prompted mobility of all metals, while the basic environment increased Mn and Cu levels and simultaneously decreased bioavailable content of Zn.

### 2.2. Effect of pH on Plant Growth and Photosynthesis

Influence of soil pH on index of chlorophyll in leaves (Chl), the activity of net photosynthesis (PN), stomatal conductance (Gs), transpiration rate (E) and intercellular concentration of CO_2_ (Ci) augmented with heights of all investigated plants are in Figure 1. The largest plant heights of basil and dandelion were observed for original soil at pH = 6.0 Lemon balm behaved in and odd way, the highest plants were grown at pH = 4.7. The basic environment (pH = 8.7) substantially inhibited growth of all three plants (Appendix A). Alterations in the height of basil and dandelion plants were consistent with their photosynthesis indicators while lemon balm behaved in a different way. The largest Chl was observed at pH = 6.0 for all investigated plants while the highest net photosynthesis was found in the dandelion grown in soil at pH = 6.0. The intercellular CO_2_ concentrations did not differ substantially for all pH values. The alkaline reaction of the soil (pH = 8.7) reduced the net photosynthesis in lemon balm by 40% as compared to the control treatment (pH = 6.0) while the acidic conditions (pH = 4.7) prompted only 36% decrease. Dandelion and basil plants reacted in a less pronounced way. The most sensitive to alkaline environment was dandelion. It reduced photosynthesis intensity by closing stomata, which further reduced transpiration and CO_2_ capture by the plant. Those processes help to keep the intercellular CO_2_ concentration at a stable level and do not affect chlorophyll content substantially. Therefore, the plant photosynthesis apparatus remains ready for increasing its efficiency upon pH returning to neutral values.

### 2.3. Effect of pH on Heavy Metals Contents in Plants

Metal contents in either above-ground parts or roots of basil, lemon balm and dandelion plants cultivated in pH modified soils are in Figure 2. Metals uptake was assessed by the one-way ANOVA at the 0.95 probability level (Table 2) and followed by the post hoc Tukey’s test [28]. The null hypothesis supported by those calculations was whether changes of soil pH affected migration of metals from soil to the plant.

In basil Mn, Cu and Zn accumulation in roots was higher than that in above-ground parts (Figure 2). In lemon balm and dandelion concentrations of manganese in above-ground parts were larger than those in roots while zinc was accumulated mostly in roots. Metals uptake by plants may be conveniently assessed by the bioaccumulation factor (BAF), translocation factors (TF) and transfer coefficients (TC) (Table 3). The former is defined as the ratio between the metal concentration in the above-ground parts of plant and its content in the soil. TF is the ratio between respective concentrations in above-ground parts and the plant roots, while TC represents a fraction of the metal content in roots as compared to that in the soil [29,30,31].

In all investigated plants increasing soil pH hampered Zn transport from soil to either roots or shoots. The manganese uptake was intensified in dandelion and lemon balm plants in either acidic or alkaline soils while the latter environment restricted Mn transport in basil. The copper uptake was prompted by acidic conditions in basil and almost neutral environment (pH = 6.0) in dandelion and lemon balm. Notably, migration of metals from roots to shoots as defined by respective translocation factors [32] is quite pH-independent. Soil pH has very limited influence on metals transfer within the plant tissues as indicated by the TF values. High metal mobility in the acidic soil stimulated their uptake by roots in basil and dandelion. However, the dandelion roots accumulated manganese most efficiently in the basic environment. Lemon balm prompted copper uptake by above-ground parts in the neutral and basic treatments.

In the latter plant, the only exception was manganese, which was better absorbed by roots in the basic environment. Lemon balm prompted copper uptake by above-ground parts in the neutral and basic treatments.

## 3. Discussion

Bioavailability of heavy metals to plants is strictly related to their mobility in soil. The latter is governed by the soil pH, which is also an important factor responsible for the metal migration in plants tissues [1]. At low pH metals are likely to exist in cationic forms, which are less prone to sorption on soil surfaces [33].

The solubility of Mn in soil solution is strongly related to pH with the lowest values in the neutral or close to neutral environments. On the other hand, acidic soils prompt the Mn solubility while anionic complexes facilitate increased Mn solubility in alkaline solutions. Formation of copper complexes strongly affects its solubility in soil. At pH below 7, copper exists as hydrolysis products (CuOH^+^ and Cu_2_(OH_2_)_2_^2+^), while above pH = 8 it is transformed to anionic hydroxy complexes.

Zinc is the most amphoteric of all investigated metals. In acidic conditions its cations are highly mobile. At neutral pH its absorption is significantly governed by exchange within the soil cation sites. The latter effect is mediated by competing cations, which decrease Zn^2+^ adsorption and increase its mobility. In alkaline environments, zinc promptly reacts with organic ligands yielding complex entities with inorganic anions.

Depending on the soil character the latter mechanism may either increase or decrease zinc mobility [34].

Distinct plants enhance metals uptake at specific pH values [35]. Particular species have different root morphologies and distributions in soil profiles. They display different patterns of nutrient uptake and H^+^ excretion along the root axis as well as down the soil profile. Additionally, plants have the ability to buffer pH changes [36].

Root uptake is mainly facilitated by low pH in all investigated plants. Exceptions were identified for uptakes of Mn in dandelion and Cu in lemon balm. A similar situation was already observed in *Valeriana officinalis* L. grown in soil at very alkaline, pH = 10.0 [37]. At high pH both metals are more prone to complexation by soil organic matter, which increases their bioavailability to plants. Plants adapt to alkaline environment by secreting acidic substances from their roots to the rhizosphere [38]. This process, which facilitates nutrients uptake, is very plant-specific [39,40]. However, final yield depends on either complexation or rhizosphere acidification and may be treated as an indicator of particular plant resistance to diverse pH conditions [34].

Cell walls of the apoplast are incrusted with sulfuric and carboxylic groups, the charges of which may be easily modified by the pH alterations. In a charged form those entities bind copper cations and decrease their transport to the upper parts of the plant. On the other hand, accumulation of copper chelates in symplast may be facilitated by their substantial affinity to amino acids, peptides and proteins.

Quite the similar effect was identified by Zhao et al. 2013 [35]. They observed that morphological parameters of P. *lactiflora* were decreased in respect to pH = 7.0 at either pH = 4.0 or 10.0. This could be triggered by toxic effect of H^+^, Mn^2+^ and Zn^2+^ cations and reduced availability of Mg, Ca, K, P and Mo elements in acidic conditions. In alkaline environment, the uptake of essential metals and other trace elements was reduced [41]. Soil pH alters metal uptake mostly at the root level. Further transport to shoots is altered to a much lesser extent. Metals enter the root through either transmembrane symplastic or apoplastic pathways [42]. The latter uses space between cells. Passing through the membrane is facilitated by transporter proteins, which are specifically related to particular elements [43]. Similar mechanisms govern further transport from roots to shoots. In all investigated plants metals migration from roots to shoots was to a much lesser extent affected by the pH modifications with copper being the most prone. Zinc and manganese are usually transported by the symplastic pathways, while copper partially follows the apoplastic avenues [44]. We speculate that the latter mechanism is more affected by soil pH changes, which propagate within plant tissues through free spaces of the apoplast. Cell walls of the apoplast are incrusted with sulfuric and carboxylic groups, the charges of which may be modified by the pH alterations. In a charged form those entities bind copper cations and decrease their transport to the upper parts of the plant. On the other hand, accumulation of copper chelates in the symplast may be facilitated by their substantial affinity to amino acids, peptides and proteins [44,45].

Divalent cations are the only Mn species that are available for plants. On the contrary, the higher valence oxides, which are the products of autooxidation at alkaline pH, are not easily accessible. Manganese uptake is a biphasic pathway. Initially, cations are reversible and rapidly absorbed by the root apoplast. Further transport is mostly realized in an active way by the intercellular symplast transport. However, as pointed out by Humphries et al. [46] and Milaleo et al. [47] the explicit mechanism has not been thoroughly characterized as yet. Zinc uptake follows the symplastic pathway and is supported by transport proteins, which are located in cell membrane. It actively depends on the plant metabolism. Cell interior is isolated from the outer space by the membrane, which helps to maintain neutral pH regardless the external environment. This situation facilitates binding zinc cations to organic chelators inside the cell and significantly reduces their mobility [48].

In all investigated plants the highest rates of photosynthesis were observed at neutral pH, while the lowest were found for alkaline environment. The intercellular CO_2_ and stomatal conductance roughly followed that pattern but were less vulnerable upon extreme pH values. Copper uptake by the lemon balm followed the above distributions of photosynthesis parameters. Additionally, in this herb net photosynthesis, transpiration rate, stomatal conductance and intercellular CO_2_ were among the lowest observed in all three investigated plants. Copper and manganese migration is related to the activation of side groups in the apoplast. This effect increased retention of metals in above-ground parts of plants. Therefore, we speculate that Mn and Cu migration in dandelion and lemon balm, respectively, was realized in a passive way through empty spaces of the apoplast. On the other hand, a systematic decrease in zinc concentrations in either roots or above-ground parts upon pH increase was observed for all investigated herbs. Its concentrations in either roots or upper parts of the plants were the highest at the lowest pH level.

The photosynthesis apparatus is located inside the cell and strongly dependent upon internal pH stabilization. Hydrogen cations enter cell cytoplasm via a number of ways, with the transporting channels being the most important for pH control. Our results indicate that photosynthetic efficiency at different pH is positively correlated with uptake of passively transported metals such as Cu and Mn. The active zinc transport is less pH-dependent.

## 4. Materials and Methods

### 4.1. Soil Analysis

Soil samples were collected on agriculture farmland in Lagiewniki (51°51′ N, 19°28′ E, which was not fertilized nor subjected to pesticide treatments for at least year before the beginning of the experiment; located away from traffic) according to the PN-ISO 10381-4: 2007 [49] in August 2019. The sampled soils were dried in an air-circulating room, followed by removal of stones and plant residues, and were passed through a 2 mm stainless steel sieve.

Soil pH was measured according to PN-ISO 10390:1997 [23]. Organic matter content in soil was determined by the gravimetric method [24,25]. For analysis of bioavailable metals in soil, air-dried soils were extracted by 0.5 mol L^−1^ of HCl according to standard [50]. The method involves shaking 10 g soil sample with 100 mL of 0.5 mol/l HCl for 1 h at room temperature (25 °C) followed by measurement of element concentrations in the filtrate.

The mixture of concentrated HNO3 (6 mL) and HCl (2 mL) was applied during mineralization of soil (0.5000 g) with the Anton Paar Multiwave 3000 closed system instrument. Metals contents were measured in soil samples by the HR-CS FAAS with the contraAA 300 (Analytic Jena spectrometer). Five replications were conducted for each sample.

### 4.2. The Soil pH Adjustment

Soil pH was modified to values 4.7 and 8.7. Soil samples (300 g) were placed in plastic pots (5 for each pH value) and then 150 mL (0.5 mol L^−1^) H_2_SO_4_ or 60 g CaO were added to each sample and carefully mixed. Afterward, the soil samples were air-dried and subjected to pH measurements. Moreover, the bioavailable forms of manganese, copper and zinc were determined in soil samples with modified pH values.

### 4.3. Cultivation of Plant Material

All herbs were cultivated under laboratory conditions by the pot method [36] from March to July 2019. Seeds of *Melissa Officinalis, Taraxacum officinalis, Ocimum basilicum* (from P.H. Legutko company, Poland) in an amount of approximately 20 seeds per pot were placed in plastic containers with a diameter of 12 cm (45 pieces −5 for each pH value and control). The cultivation was conducted in green house under controlled conditions: the temperatures were 23 ± 2 and 16 ± 2 °C for day and night, respectively; the relative humidity was limited to 70–75%; and the photosynthetic active radiation (PAR) during the 16 h photoperiod was restricted to 400 μmol m^−2^ s^−1^. All plants were regularly watered with deionized water. After 3 months the herbs were harvested separately into above-ground material and roots. The weights of the individual parts of the plants were determined. The roots were washed in tap water and deionized water. The harvest was oven-dried at 45 °C to a constant weight, homogenized and ground.

### 4.4. The Morphological and Physiological Parameters of Plants

The above-ground parts lengths were determined by measurement of distance between the crown and the leaf tip (cm). Index of chlorophyll was measured using the Konica Minolta SPAD-502Plus, Tokyo, Japan. Activity of net photosynthesis (PN), stomatal conductance (Gs), intercellular concentration of CO_2_ (Ci) and transpiration (E) were determined with the gas analyzer apparatus CIRAS-3 (Portable Photosynthesis System, Amesbury, MA, USA) [51,52]. All measurements were made five times on separate plants without damaging the plant material.

### 4.5. Determination of Heavy Metals in Herbs

The concentrations of metals in roots and above-ground parts of herbs were determined in mineralizates of each plant material. The same protocol as used for soil analysis was applied.

Quality assurance and quality control (QA/QC) for metals in plants samples were estimated by determining metal contents in the certified reference material INCT-MPH-2 containing a mixture of selected Polish herbs [53] (Appendix A).

## 5. Conclusions

In summary, soil pH alters biological parameters as well as manganese, copper and zinc uptake by basil, dandelion and lemon balm in a number of ways. This issue is particularly important for keeping medicinal value of herbal crops at a high and stable level. Investigated metals play significant role in the photosynthesis process. Its efficiency at different soil pH values positively correlates with uptake of Cu and Mn, while Zn behaved in a more unpredicted way. In all investigated plants the highest rates of photosynthesis were observed at neutral pH, while the lowest were found for alkaline environment. The intercellular CO_2_ and stomatal conductance were less vulnerable upon extreme pH values. The most sensitive to alkaline conditions was dandelion. It reduced photosynthesis intensity by closing stomata, which further limited transpiration and CO_2_ capture by the plant. Those processes help to keep intercellular CO_2_ concentration at stable level without affecting chlorophyll content substantially.

## Figures and Tables

**Figure 1 molecules-27-04671-f001:**
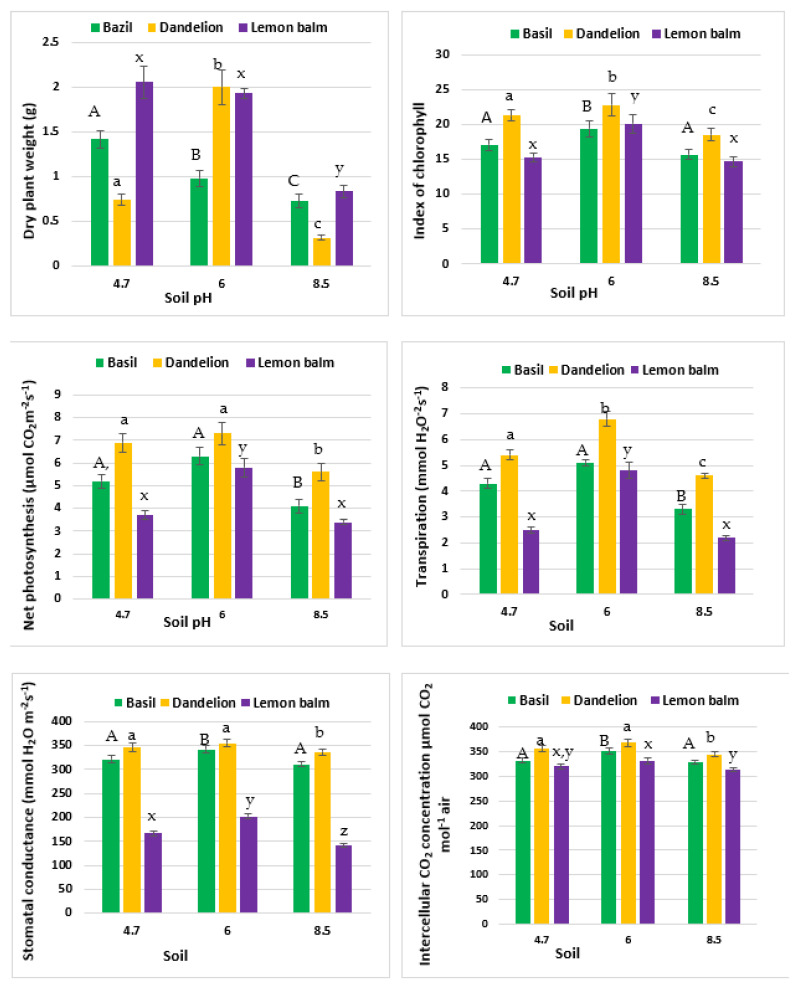
The effect of soil pH on the height of plants, index of chlorophyll, net photosynthesis, stomatal conductance, transpiration rate and intercellular concentration of CO_2_. Specific letters illustrate the statistically significant differences as computed with the Tukey’s HSD test (*p* = 0.95).

**Figure 2 molecules-27-04671-f002:**
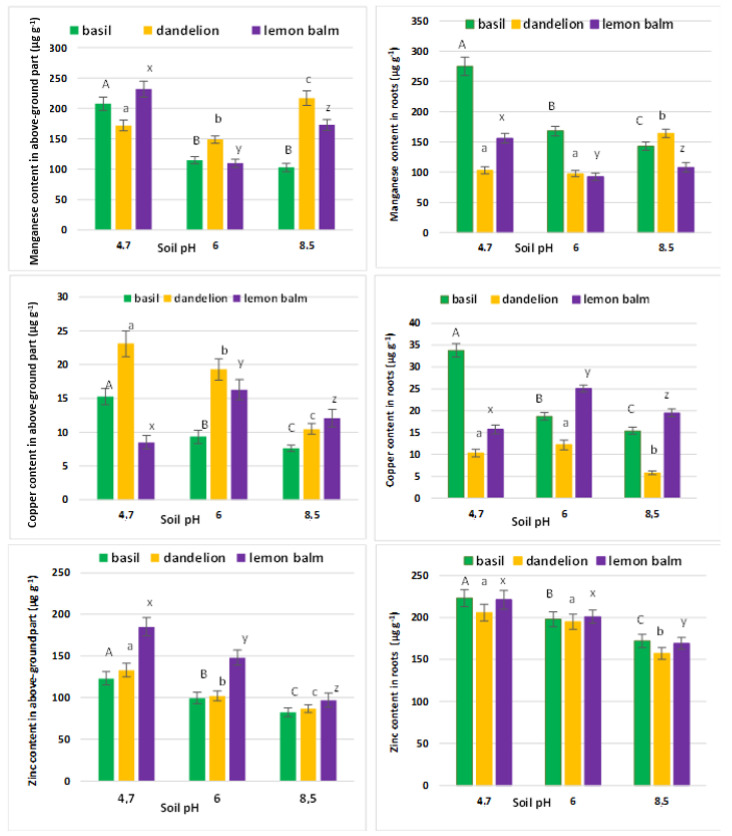
Manganese, copper and contents in above-ground parts and roots of basil, dandelion and lemon balm plants displayed against the pH modifications. Specific letters illustrate the statistically significant differences as computed with the Tukey’s HSD test (*p* = 0.95); roots and above-ground parts are treated independently.

**Table 1 molecules-27-04671-t001:** Analysis of soil without additives (**a**). Bioavailable forms of Mn, Cu, Zn in either initial soil without additives (pH = 6.0) or modified soil samples with pH adjusted to 4.7 and 8.7, respectively (**b**). Specific letters illustrate the statistically significant differences as computed with the Tukey’s HSD test (*p* = 0.95).

(a)				
**pH**		6.0		
**Organic matter**		32.5%		
**Metal concentration (µg·g^−1^)**		**Mn**	**Cu**	**Zn**
**Total forms**		198 ± 5	25.6 ± 0.6	201 ± 4
**(b)**				
**Bioavailable forms**	**pH = 4.7**	133 ± 1.7 ^A^	14.6 ± 0.7 ^D^	136 ± 6 ^F^
	**pH = 6.0** **control**	98.7 ± 1.8 ^B^	12.2 ± 0.7 ^E^	117 ± 2 ^G^
	**pH = 8.7**	119 ± 2 ^C^	15.4 ± 0.5 ^D^	91.8 ± 0.9 ^H^

**Table 2 molecules-27-04671-t002:** The one-way ANOVA for manganese, copper, zinc contents in plant across the soil pH. Critical Snedecor’s F value is Fcryt = 3.88533.

	Basil		Dandelion		Lemon Balm	
	Above-Ground Parts	Roots	Above-Ground Parts	Roots	Above-Ground Parts	Roots
**Mn**	*p* = 8.17 × 10^−10^ F = 190.2278	*p* = 1.37 × 10^−9^ F = 174.0622	*p* = 7.47 × 10^−7^ F = 56.9940	*p* = 6.10 × 10^−9^ F = 134.3608	*p* = 2.59 × 10^−9^ F = 155.8941	*p* = 8.43 × 10^−8^ F = 84.6094
**Cu**	*p* = 1.02 × 10^−7^ F = 81.8463	*p* = 5.61 × 10^−11^ F = 300.6741	*p* = 1.56 × 10^−7^ F = 75.7392	*p* = 5.96 × 10^−7^ F = 59.3978	*p* = 6.60 × 10^−6^ F = 37.8033	*p* = 1.60 × 10^−8^ F = 113.4664
**Zn**	*p* = 9.68 × 10^−6^ F = 35.0993	*p* = 1.80 × 10^−5^ F = 31.0742	*p* = 1.07 × 10^−6^ F = 53.3690	*p* = 1.14 × 10^−5^ F = 33.9830	*p* = 7.33 × 10^−8^ F = 86.7552	*p* = 1.06 × 10^−5^ F = 34.4554

**Table 3 molecules-27-04671-t003:** Bioaccumulation factors (BAF), translocation factors (TF) and transfer coefficients (TC) calculated for basil, lemon balm and dandelion, respectively. Elements are shown in decreasing order of particular factor. Each value is the average of the data from five replicates.

	BAF	TF	TC
pH	Basil		
**4.7**	**Mn** (1.05) > **Zn** (0.61) > **Cu** (0.60)	**Mn** (0.76) > **Zn** (0.55) > **Cu** (0.45)	**Mn** (1.39) > **Cu** (1.32) > **Zn** (1.11)
**6.0 control**	**Mn** (0.58) > **Zn** (0.50) > **Cu** (0.36)	**Mn** (0.68) > **Zn** (0.50) = **Cu** (0.50)	**Zn** (0.99) > **Mn** (0.85) > **Cu** (0.73)
**8.5**	**Mn** (0.52) > **Zn** (0.41) > **Cu** (0.30)	**Mn** (0.72) > **Zn** (0.48) > **Cu** (0.47)	**Zn** (0.86) > **Mn** (0.72) > **Cu** (0.60)
**Dandelion**			
**4.7**	**Mn** (0.87) > **Zn** (0.61) > **Cu** (0.39)	**Cu** (2.24) > **Mn** (1.67) > **Zn** (0.65)	**Zn** (1.02) > **Mn** (0.52) > **Cu** (0.40)
**6.0 control**	**Mn** (0.75) > **Zn** (0.51) > **Cu** (0.47)	**Cu** (1.58) > **Mn** (1.53) > **Zn** (0.52)	**Zn** (0.97) > **Mn** (0.49) > **Cu** (0.48)
**8.5**	**Mn** (1.10) > **Cu** (0.26) > **Zn** (0.25)	**Cu** (1.83) > **Mn** (1.32) > **Zn** (0.55)	**Mn** (0.83) > **Zn** (0.78) > **Cu** (0.22)
**Lemon balm**			
**4.7**	**Mn** (1.17) > **Zn** (0.92) > **Cu** (0.33)	**Mn** (1.49) > **Zn** (0.84) > **Cu** (0.54)	**Zn** (1.10) > **Mn** (0.79) > **Cu** (0.61)
**6.0 control**	**Zn** (0.74) > **Cu** (0.64) > **Mn** (0.56)	**Mn** (1.19) > **Zn** (0.74) > **Cu** (0.65)	**Zn** (1.00) > **Cu** (0.98) > **Mn** (0.47)
**8.5**	**Mn** (1.06) > **Zn** (0.48) > **Cu** (0.47)	**Mn** (1.60) > **Zn** (0.57) > **Cu** (0.62)	**Zn** (0.84) > **Cu** (0.76) > **Mn** (0.55)

## Data Availability

Not applicable.

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
