# Peer review of "The Impact of Soil pH on Heavy Metals Uptake and Photosynthesis Efficiency in *Melissa officinalis*, *Taraxacum officinalis*, *Ocimum basilicum"

_molecules, 2022, doi:10.3390/molecules27154671_

Round 1

Reviewer 1 Report

This paper focused on the impact of soil pH on heavy metals uptake and will surpply important information for the heavy metals phytoremediation. However, the language is hard to read. the tables and figures need add the significant analysis. and the results need to be improved for professional and articulate.

Reviewer 2 Report

In your paper “The impact of soil pH on heavy metals uptake by herbs. The Melissa Officinalis, Taraxacum officinalis, Ocimum basilicum case study” you report results from a greenhouse experiment with 3 different herbs used for medical applications focussing on the effects of soil pH on photosynthesis parameters, growth and heavy metal uptake.

To my understanding your manuscript does not really fit to the scope of Molecules, as this journal is focussing on the science of chemistry and interfacing disciplines, while you report results from a pot experiments focusing on the influence of soil pH on heavy metal uptake by plants. Consequently, later on most of your interpretation in the Discussion section is based on plant physiological aspects. Having the scope of Molecules in mind, I would have expected that you will show much more details on different binding forms of the heavy metals in soil and in the plant tissue.

Furthermore, your manuscript has many other shortcomings. As far as I can judge, your manuscript does not really give substantial new insight into the subject because it fails to report consolidated data and to really work out underlying effects.

Although you obviously follow the “Guidelines for the Authors” the overall structure of the paper is not according to international standards for scientific journal (M&M section should follow directly after the Introduction).

The title of the paper is unclear and does not really enable the reader to get an idea about your experiment. In fact you have done a pot experiment where you have considerable differentiated soil pH at the start of the experiment, which has many short and medium effects on other soil characteristics [e.g. microbial activity] and which certainly is not comparable to soil pH changes under field conditions which normally take some time).

Keywords should not repeat terms already given in the Title!

The abstract neither is written in a way to encourage the reader to read the whole paper nor to really understand the story (the methodological approach remains unclear; names of analytical instruments should not be mentioned; obtained results are not clearly presented; a concluding sentences is missing).

Some parts of the Introduction are rather unfocused and not directly related to the topic of the manuscript (e.g. lines 58-62). A scientific relevant hypothesis at the end of the Introduction is missing.

Results section

Data in Tab. 2 should be split-up into 2 tables. It remains unclear why you analysed more “bioavailable” (no clear definition is available in the context of the Tab. 2) Mn and Cu under alkaline soil conditions.

Figure 1: To my understanding, the setup of the graphs is not meaningful. In fact, you want to show the influence of soil pH on parameter “x”. Therefore, for each plant all 3 pH levels should be grouped! Furthermore, the statistical details are missing! “Plant height” is not a scientifically relevant parameter (I would have expected that you show data on aboveground biomass).

Why do you show results on “stomatal conductance” and “intercellular CO2 concentration”?

Table 3: These statistical details are not relevant! You must clearly indicate in the Figures which “pH treatments” are “statistically significant different”!

Figure 2: see comment for Fig. 1 concerning “grouping of the pH treatments”. You should use always the same colour code for the 3 plant species. You should use the same Y axis spread for each of the 3 heavy metals (e.g. Cu y-axis for above ground an roots from 0 – 40 µg g-1). Do not use the term “content” but “concentration”.

Table 3 is really hard to read and understand!!!

The Discussion section is not really interpreting the results from the pot experiment. In fact you mostly use well-known plant physiological arguments/basic knowledge, but you have not done any measurement in your experiment (e.g. L 160-164, L178-182, ….).

L145-146: Why are cationic forms of heavy metals less prone to sorption on soil surfaces?

L147-150: Any reference for this statement?

L154-155: add reference!

L197-203: repeating information from the results section.

L206ff: I do not get the point of your argumentation!

M&M section

L215: “Row soil”????? (even “raw soil” would not be an acceptable wording!!!!)

L217: “use of pesticides” relevance?

L222: I have seen several definitions for “bioavailable” heavy metals: What are the reasons to use the extraction with “0.5 mol L−1 of HCl”? Add some more details (e.g. soil:extract ratio, duration)

L231: What do you mean with “adequate amounts of (0.5 mol L-1) H2SO4 or CaO were added”. Here you MUST explain with much more details how you modified the soil pH!

L235-253: move to Introduction

L255: What do you mean with “pot method”?

L223-227 and 276-280. Repeating information

L280-283 and Tab. S1: relevance?

References

Relevance of ref. 47 – ref. 57?

Rather too many references for such a rather “simple” story.

Several mistakes concerning spelling, formatting, etc.

Although I am not a native English speaker, I consider the language quality of the text is not adequate for an international journal.

Based on these arguments I recommend that this paper is not acceptable for publication in its present form.
